# Sleep Architecture Changes in Diabetes

**DOI:** 10.3390/jcm13226851

**Published:** 2024-11-14

**Authors:** Yuanjie Mao

**Affiliations:** 1Diabetes Institute, Ohio University Heritage College of Osteopathic Medicine, Ohio University, Athens, OH 45701, USA; ymao@ohio.edu; Tel.: +740-593-2396; Fax: +740-593-1342; 2Endocrinology & Diabetes Clinic, OhioHealth Castrop Health Center, Athens, OH 45701, USA

**Keywords:** diabetes mellitus, hyperglycemia, sleep architecture, sleep alternation

## Abstract

Data on the relationship between sleep architecture and diabetes are limited. However, some evidence suggests that slow-wave sleep (SWS) plays a crucial role in maintaining normal glucose homeostasis and influences insulin secretion capacity. Diabetes is often associated with reduced SWS, even in the absence of sleep-disordered breathing. Notably, selective suppression of SWS—without reducing total sleep time—can lead to significant increases in insulin resistance, decreased glucose tolerance, and a higher risk of diabetes. Given the growing interest in non-pharmacological lifestyle interventions, such as modifying sleep architecture, it is important to understand how sleep patterns differ in individuals with diabetes and whether these alterations impact diabetes risk and glycemic control. This review aims to provide a concise overview of the current findings on sleep architecture changes in people with diabetes.

## 1. Introduction

Diabetes mellitus is one of the most prevalent chronic diseases, affecting 10.5% of the adult population aged 20–79 years and approximately 537 million people worldwide in 2021 [1]. This condition, characterized by glycemic dysregulation, leads to a reduced quality of life, numerous diabetic complications, and early mortality. People with diabetes often experience insufficient sleep duration and impaired sleep quality [2,3]. Additionally, sleep-disordered breathing, particularly obstructive sleep apnea (OSA), is common among those with diabetes [4,5]. Recent studies have also found associations between poor subjective sleep and worsening glycemic control, as indicated by high hemoglobin A1c levels [6,7,8].

Sleep is a dynamic process characterized by distinctive stages that progress in a cycle throughout the night. The duration and characteristics of each stage and cycle make up the sleep architecture [9]. The two primary stages of sleep are rapid eye movement (REM) and non-rapid eye movement (NREM) sleep. According to the American Academy of Sleep Medicine, there are four recognized sleep stages based on electroencephalographic, electromyographic, and electrooculographic measures: stage 1 NREM (N1), a brief transition between wake and sleep; stage 2 NREM (N2), characterized by high-amplitude K complexes and spindles; stage 3 NREM (N3), known as slow-wave sleep (SWS), featuring slow oscillations; and stage 4 rapid eye movement (REM), often referred to as dream sleep, marked by rapid eye movements and theta activity. The onset of sleep begins with wakefulness and progresses rapidly to REM sleep by approximately 60 to 90 min. This is followed by cyclical alterations between REM and NREM every 90 to 120 min until awakening. Most REM sleep occurs during the second half of the night, with NREM sleep dominating the first half of the night.

Data on sleep architecture changes and diabetes are limited. SWS appears to play a crucial role in maintaining normal glucose homeostasis. During SWS, cerebral glucose utilization decreases while plasma glucose levels rise [10,11]. SWS duration is related to beta-cell function and the capability for insulin secretion [12]. Selective suppression of SWS, without altering total sleep time, significantly increases insulin resistance, leading to reduced glucose tolerance and a high risk of diabetes [13]. Additionally, studies have found a negative association between REM sleep duration and energy balance [14]. The associations of these sleep architecture changes with diabetes warrant further studies. Importantly, sleep architecture interventions are emerging as popular non-pharmacological strategies for improving health, as evidenced by recent clinical trials [15]. This review aims to succinctly discuss the current findings on sleep architecture changes in people with diabetes.

## 2. Sleep Architecture Changes in Diabetes

Studies have found a U-shaped relationship between total sleep duration and both short- and long-term glycemia. Optimal glucose metabolism is observed when individuals achieve 7.5 to 8.5 h of sleep per night [12]. This is consistent with adult data noting U-shaped associations between self-reported sleep duration and incident diabetes risk [16]. Beyond total sleep duration, changes in sleep architecture, particularly the role of SWS and REM, have been explored in relation to diabetes risk.

### 2.1. SWS Changes in Diabetes

SWS, as the deeper stage of NREM sleep, is thought to be the most “restorative” sleep. There is indeed evidence that SWS plays a crucial role in waking neurobehavioral function, particularly in memory consolidation [13]. The initiation of SWS is temporally associated with transient metabolic, hormonal, and neurophysiologic changes, all of which could potentially affect glucose homeostasis. These include decreased brain glucose utilization, stimulation of growth hormone release, inhibition of corticotropic activity, decreased sympathetic nervous activity, and increased vagal tone [13]. Previous studies reported positive associations between SWS and beta-cell function and insulin secretory measures after adjustments for potential confounders such as sex, pubertal stage, body weight, and OSA [12]. This study suggests that at least one hour of SWS may be required for stable insulin secretion. Increasing SWS duration could further enhance insulin secretion capability [12]. On the other hand, SWS loss can decrease insulin secretion and increase insulin resistance [13]. These results may explain the association between SWS duration and diabetes risk.

Adults with diabetes have been reported to have shorter SWS durations than nondiabetic adults (3.9 ± 6.0% vs. 8.4 ± 4.6%; *p* = 0.012; n = 44; 8 women, 36 men, aged 57.5 ± 5.5 years, BMI 33.8 ± 5.9 kg/m^2^, AHI 29.6 ± 22.2 episodes/h) matched individually for age, sex, body mass index (BMI), smoking, and apnea–hypopnea index (AHI), one of the markers for OSA [17]. Consistently, in a cross-sectional study including 2026 participants (mean age of 69 years and diabetes prevalence of 28%), those with a high SWS proportion were less likely to have prevalent diabetes [18]. In the following prospective analysis of 1251 participants and 129 incident cases over 6346 person-years of follow-up, a curvilinear relationship was observed between a higher SWS proportion or longer SWS duration and lower incident diabetes risk [18].

However, there are inconsistent reports. Some studies found that the SWS duration and percentage were greater among people with diabetes (+7.6 min, 95%CI 0.6, 14.6, and +2.4%, 95%CI 0.6, 4.2) in a cross-sectional sample with 1074 participants with AHI < 30, including 113 people with diabetes [19], and other studies found no difference among those with diabetes compared to those without diabetes [20,21]. These discrepancies may stem from the use of varying methodologies across studies, such as differences in sample sizes, participant demographics (including age and body weight), and covariate adjustments, particularly regarding sleep-disordered breathing. Most of these studies are small-sized and cross-sectional in nature. Most importantly, all recent studies have been focused on the duration and quantity aspects of sleep architecture, and none of them have investigated sleep efficiency and sleep depth. Overall, while the relationship between SWS and diabetes risk is supported by many studies, further research is needed to clarify inconsistencies and better understand the role of SWS in metabolic health.

### 2.2. REM Changes in Diabetes

REM sleep is prevalent in the later periods of the night and is characterized by theta activity. The precise role of REM sleep is currently debated, but REM sleep has been implicated in diabetes and obesity. A pediatric study found a negative association between REM sleep duration and obesity measurements [14]. Further, in the Baependi Heart Study, a family-based cohort of adults in Brazil with about 1074 participants underwent at-home polysomnography; people with diabetes and prediabetes had less REM sleep than those without diabetes after considering potential confounders, including age, sex, BMI, and AHI [19]. Another large cohort study in the U.S. including 5874 participants also reported a decrease in REM percentage among individuals with diabetes (19.0% among diabetic vs. 20.1% among nondiabetic subjects, *p* < 0.001) after adjustment for age, sex, BMI, race, and neck circumference [20]. However, other studies have found either increased REM percentage [17] or no significant differences in individuals with diabetes [21,22]. These conflicting results may arise from differences in the study populations, sample sizes, or methodologies used to assess sleep stages like SWS changes.

### 2.3. Other Changes in Diabetes

N2 sleep, the light sleep stage, has also been linked to insulin sensitivity. One study found a positive association between N2 duration and insulin sensitivity, though the role of obesity complicates this relationship [12]. N2 duration was also negatively associated with at least one marker of OSA, including the AHI, indicating possible confounding; however, the AHI did not consistently correlate with insulin sensitivity measures. Therefore, the observed association between N2 sleep and insulin sensitivity may represent an intrinsic relationship [12].

Sleep quality and sleep efficiency could also be affected by diabetes. A study found that diabetes is associated with an altered sleep structure, with different effects according to REM (increase in nocturnal hypoxia) or NREM (increase in sleep fragmentation, also called microarousals) sleep [21]. Nocturnal hypoxia refers to having lower oxygen levels during sleep, while microarousals are brief awakenings that can disrupt sleep continuity. Sleep variables related to oxygen saturation measures, such as the percentage of time spent with oxygen saturation ≤90%, were significantly greater during the REM stage in people with type 2 diabetes [20.3 (total range 0–99.2) vs. 10.5 (total range 0–94.0)%] than that in those without diabetes. This pattern was maintained in the subgroup of patients matched by AHI [21]. In addition, people with type 2 diabetes had more microarousal events during sleep than the control subjects. These differences were mainly observed during the NREM stage [21]. Overall, while changes in REM and NREM sleep patterns are commonly observed in individuals with diabetes, the precise nature of these changes remains unclear, underscoring the need for further research to disentangle these complex relationships.

## 3. Possible Mechanisms

There are several proposed physiological mechanisms by which diabetes may disrupt normal sleep architecture, impacting both sleep quality and overall metabolic health. Diabetes increases sleep fragmentation through higher rates of microarousals (brief awakenings during sleep) during non-REM sleep and favors intermittent hypoxia (periods of low oxygen levels) during REM sleep [21]. Both sleep fragmentation and intermittent hypoxia are the main mechanisms by which sleep breathing disorders exert their negative metabolic, hemodynamic, inflammatory, and vascular effects [21].

One of the other potential mechanisms beyond the presence of sleep-disordered breathing is inflammation. It is well known that diabetes causes chronic inflammation [22]. Chronic inflammation is linked to alterations in sleep in diabetes, with higher levels of inflammatory markers like IL-6 being associated with longer REM latency (the time between sleep onset and the first REM stage) and a lower REM proportion [23,24]. Higher levels of proinflammatory makers are also associated with longer SWS duration [23,24]. Therefore, inflammation might play a role in glycemic regulation and the development of diabetes.

The microvascular complications of diabetes are long-term complications that affect small blood vessels. These typically include retinopathy, nephropathy, and neuropathy. Microvascular complications in diabetes have been associated with decreased REM duration and poor sleep efficiency [25]. Neuropathic pain from diabetic neuropathy has a strong association with sleep disturbance [26].

Moreover, people with type 2 diabetes showed reduced parasympathetic activity but preserved short-term sympathetic function compared to the controls, indicating autonomic dysfunction with predominantly parasympathetic impairment [27]. Meanwhile, parasympathetic activity, which is increased in SWS, stimulates glucose-induced insulin secretion [28,29]. Further, SWS plays a key role in modulating endocrine function. It suppresses the release of hormones from the hypothalamic–pituitary–adrenal axis, such as cortisol, while promoting the increased secretion of growth hormone and prolactin [30]. Enhancing SWS in healthy individuals through hypnotic suggestions significantly increased the release of growth hormone and prolactin while also shifting the sympathovagal balance toward reduced sympathetic dominance [31]. Growth hormone secretion occurs largely during SWS [32], and growth hormone may increase insulin resistance [33]. However, one available study showed that insulin-like growth factor-1 levels, the downstream marker of growth hormone, did not correlate significantly with either sleep duration or any individual sleep stage [12].

Lastly, there is a possibility that sleep requirements are altered in diabetes, leading to greater sleep pressure [19]. SWS is homeostatically controlled, with increased sleep deprivation and the associated greater sleep pressure resulting in a high demand for SWS [34]. Normally, SWS occurs primarily in the first half of the night, while REM occurs primarily in the second half of the night. In response to sleep deprivation, the body prioritizes SWS (the deep, restorative sleep stage) over REM sleep, which may explain why individuals with diabetes, who often report shorter sleep durations, have increased SWS duration and decreased REM sleep. However, sleep efficiency and sleep depth in SWS might also be affected. Future studies measuring sleep depth in SWS in people with diabetes may help with understanding whether these alterations have implications for diabetes management. The potential main mechanisms between SWS disruption and diabetes are showed in Figure 1.

## 4. Sleep Architecture Alteration and Diabetes

Recent research suggests that altering sleep architecture can influence glycemic control [35,36]. Spectral power, a measure of brain wave activity during sleep, can be increased to enhance the quality and duration of specific sleep stages, potentially leading to therapeutic benefits [37,38]. Sleep stage manipulation has typically been achieved using acoustic stimulation. Acoustic stimulation typically involves playing sounds at increasing intensities until they provoke microarousals, which can shift the participant’s sleep stage. This method has been explored to manipulate sleep architecture in clinical settings. Studies examining sleep duration and glycemic control suggest that both shorter- and longer-than-average sleep durations are associated with poorer glycemic control [39]. However, the specific role of different sleep stages, such as REM and SWS, in regulating glycemia is still an emerging area of research [19]. As research in this area develops, understanding how sleep stage manipulation can improve glycemic control could result in new, non-invasive therapeutic approaches for managing diabetes.

### 4.1. SWS Manipulation

SWS is prevalent in the earlier periods of the night and is characterized by slow oscillations—synchronized brain wave activity that alternates between states of high neuronal activity (depolarization) and low activity (hyperpolarization) [40]. SWS has many restorative functions and is linked with improved metabolic and psychological functioning [41]. SWS manipulation includes SWS disruption and SWS enhancement.

The impact of SWS disruption involves acoustic auditory disruption coinciding with SWS throughout the night [42,43]. Meta-analyses found that SWS disruption led to increased insulin resistance but did not significantly alter post-prandial glucose or insulin levels in most trials [44]. Seven trials, including a total of 84 participants (73.8% male), were analyzed. Most participants were healthy, though eight had type 1 diabetes and were undergoing insulin pump therapy [44]. Some trials disrupted SWS for just one night, while others involved multiple nights [44]. Three experimental studies that suppressed SWS in healthy young adults observed impairments in glucose metabolism and insulin sensitivity [13,42,45]. For example, one trial using the intravenous glucose tolerance test reported significantly decreased glucose tolerance and insulin sensitivity after SWS disruption compared to the control conditions, though there were no significant changes in insulin response during the tolerance tests [13]. In contrast, four trials that explored the impact of SWS disruption on post-prandial glucose and insulin using oral glucose tolerance or mixed-meal tolerance tests found no significant post-prandial responses compared to the control [42,43,45,46]. Two other trials, which measured glucose disposal through insulin clamping, also reported no significant differences in post-prandial glucose and insulin levels compared to the controls [44].

Notably, although most trials did not show significant responses through post-prandial glucose or insulin levels, higher levels of SWS disruption appeared to have a dose-dependent effect on insulin resistance and post-prandial glucose levels, with larger disruptions leading to more significant changes [44]. Meta-analyses indicate that trials with larger percentage decreases in SWS resulted in more pronounced alterations in post-prandial glucose and insulin levels. For instance, reductions in SWS of 49% and 79% led to greater changes in post-prandial glucose compared to reductions of 15% and 39% [44]. Similar trends were observed for post-prandial insulin and insulin resistance [44]. One trial that reduced SWS by 87% reported significant alterations in glycemic control, while another study found that a 79% reduction in SWS significantly impacted AUC glucose and insulin levels [44]. AUC (Area Under the Curve) glucose measures the total glucose exposure over time after a meal. These findings suggest that optimizing SWS disruption methods may be crucial for understanding the relationship between SWS disruption and glycemic control.

Recently, researchers have explored methods of extending SWS activity through auditory closed-loop stimulations. This technique uses a specific type of sound known as pink noise—soft, rhythmic sounds that synchronize with the brain’s slow oscillations during SWS. The goal is to prolong SWS by aligning brain activity with the auditory stimulus [37,41,47]. One trial tested this method with 15 healthy male participants (aged 19–34) who had a normal BMI (19.2–25.0 kg/m^2^) and regular sleep/wake patterns. The participants underwent one night of SWS enhancement and one night of sham stimulation [37]. During the enhancement night, an auditory closed-loop stimulation, using clips of pink noise, was applied for 210 min. This stimulation was triggered by detecting negative half-waves in slow oscillations. However, the trial found no significant changes in post-prandial glucose or insulin levels compared to the control conditions. It is important to note that this study had limitations, including a short duration of intervention and a sample composed only of healthy individuals [37]. Future research should focus on refining SWS disruption and enhancement techniques, particularly in people with diabetes, to establish more definitive links between SWS changes and glycemic control. This could pave the way for innovative, non-invasive therapeutic strategies.

### 4.2. REM Manipulation

REM sleep, which is most prevalent in the later stages of the night, is characterized by theta activity. Both the REM and SWS stages have been linked to glycemic control [48]. Circadian misalignment, which affects REM sleep duration, also influences glycemic control [49,50]. This suggests a potential mechanism connecting REM sleep with glycemic control.

Experimental studies have shown that sleep restriction, which reduces REM sleep, is associated with a positive energy balance that could increase the risk of weight gain [51]. One available trial investigating the effects of REM sleep disruption on glycemic control used auditory disruptions during REM sleep and reported no significant changes in post-prandial glucose, insulin levels, or AUC glucose and insulin compared to the control conditions [45]. However, this study is also limited by being only a one-night intervention and using a sample composed only of healthy individuals. In addition, the decrease in REM sleep was less pronounced compared with the decrease in SWS [45].

## 5. Conclusions

As an emerging area of research that has garnered significant interest but still lacks sufficient clinical evidence for definitive conclusions, further studies are needed to explore sleep biosignatures in individuals with diabetes and the potential for sleep interventions in diabetes care and glycemic management. This article provides a concise narrative review, summarizing the currently available evidence and aiming to identify research gaps for future exploration.

In conclusion, diabetes presents a distinctive sleep biosignature with varying effects on REM and non-REM sleep stages. Increasing evidence indicates that disruptions in sleep architecture are linked to impaired glucose homeostasis, reduced insulin secretion, and a heightened risk of diabetes. Most studies suggest that individuals with diabetes tend to experience reduced REM sleep and less SWS, even in the absence of moderate to severe sleep apnea. While the exact mechanisms driving these changes remain unclear, some potential links have been proposed. Notably, selective suppression of SWS—without affecting total sleep time—can lead to significant increases in insulin resistance, reductions in glucose tolerance, and an elevated risk of diabetes [13]. Although the only available study on SWS enhancement showed no beneficial effects on diabetic parameters, it was limited by the short duration of the intervention and its focus on healthy participants.

Given the critical role of tight glycemic and weight control in diabetes management, understanding and optimizing sleep architecture may be particularly important for individuals with diabetes. Addressing sleep architecture disturbances in diabetes management could fill a significant clinical gap, highlighting the need for screening by endocrinologists and primary-care physicians. As a non-pharmacological lifestyle intervention, improving sleep architecture could be a cost-effective approach to preventing diabetes and supporting glycemic control in individuals already diagnosed with the condition.

## 6. Future Directions

Many questions remain unanswered and should be addressed in future studies. First, the long-term impact of changes in sleep architecture on diabetes is unknown. Further research is needed to determine whether reduced REM or SWS in patients with diabetes is associated with poor long-term glucose control and the development of diabetic complications. Second, more studies are required to explore the mechanisms underlying SWS/REM sleep changes in diabetes and to assess whether these mechanisms influence diabetes management or habitual sleep patterns. Third, recent studies primarily focus on the duration and proportion aspects of sleep architecture. More data are needed to examine the impact of sleep depth and sleep efficiency in different sleep stages on diabetes. Fourth, the therapeutic effects of altering sleep architecture remain unclear. Future research should evaluate the potential of modifying sleep architecture for diabetes management, particularly through long-term, large-scale clinical studies focused on enhancing sleep architecture. Fifth, future studies should investigate changes in sleep architecture across different types and stages of diabetes. Since each type and stage may present specific features, they could influence outcomes differently. Research should also explore the preventive effects of altering sleep architecture in prediabetes, as well as its therapeutic potential in type 1 and secondary diabetes. Lastly, future efforts should aim to unravel the complex interactions between glucose homeostasis and sleep integrity. Studies should assess whether current diabetes treatments, such as medications and insulin pumps, can improve sleep architecture.

## Figures and Tables

**Figure 1 jcm-13-06851-f001:**
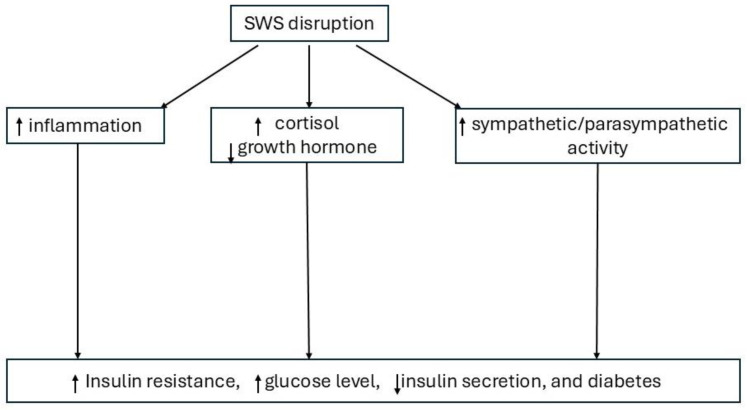
Potential main mechanisms of SWS disruption resulting in diabetes. SWS: slow-wave sleep.

## Data Availability

Not applicable.

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
