# Peer review of "Sleep Architecture Changes in Diabetes"

_jcm, 2024, doi:10.3390/jcm13226851_

Round 1

Reviewer 1 Report

Comments and Suggestions for Authors

Is this a review article? if yes, please add your search terms, search database, search strategies, and data collection, and screening process. And if this is a narrative review or other types of review? 

when discussing the discrepancies across studies, it would benefit from more detailed information on specific methodologies contributing to inconsistent findings (e.g., sample size? study population characteristics? or various sleep measurements? )

To help readers understand the unique contributions of this review, consider adding a section in the discussion outlining what uniqueness of your work can contribute to the fields. 

Author Response

Comments 1: Is this a review article? if yes, please add your search terms, search database, search strategies, and data collection, and screening process. And if this is a narrative review or other types of review?

Response: Thank you very much for your comment. This is just a narrative review, not a systemic review or meta-analysis.

Comments 2: When discussing the discrepancies across studies, it would benefit from more detailed information on specific methodologies contributing to inconsistent findings (e.g., sample size? study population characteristics? or various sleep measurements?)

Response: Thank you very much for this helpful comment.

Based on this comment, we have added more detailed information on the specific methodologies on page 2 lines 78-82 in the revised manuscript as below.

Before: “Adults with diabetes have been reported to have shorter SWS duration than non-diabetic adults (3.9 +/- 6.0% vs 8.4 +/- 4.6%; P = 0.012;) after adjustment for age, sex, body mass index (BMI), smoking, and apnea-hypopnea index (AHI), one of the markers for OSA [17].”

After: “Adults with diabetes have been reported to have shorter SWS duration than non-diabetic adults (3.9 +/- 6.0% vs 8.4 +/- 4.6%; P = 0.012; n = 44; 8 women, 36 men, aged 57.5 +/- 5.5 years, BMI 33.8 +/- 5.9 kg/m2, AHI 29.6 +/- 22.2 episodes/hr) matched individually for age, sex, body mass index (BMI), smoking, and apnea-hypopnea index (AHI), one of the markers for OSA [17].”

On page 2 lines 82-84 in the revised manuscript as below.

Before: “Consistently, in a cross-sectional study including 2026 participants, high SWS proportion were less likely to have prevalent diabetes [18].”

After: “Consistently, in a cross-sectional study including 2026 participants (mean age of 69 years and diabetes prevalence of 28%), high SWS proportion were less likely to have prevalent diabetes [18].”

On page 2 lines 88-92 in the revised manuscript as below.

Before: “However, there are inconsistent reports. Some studies found that SWS duration was greater among people with diabetes [19] and other studies found no difference among those with diabetes [20, 21].”

After: “However, there are inconsistent reports. Some studies found that SWS duration and percentage were greater among people with diabetes (+7.6 min, 95%CI 0.6, 14.6, and +2.4%, 95%CI 0.6, 4.2) in a cross-sectional sample with 1074 participants with AHI <30 including 113 with diabetes [19] and other studies found no difference among those with diabetes compared to those without diabetes [20, 21].”

Comments 3: To help readers understand the unique contributions of this review, consider adding a section in the discussion outlining what uniqueness of your work can contribute to the fields.

Response: Thank you very much for this comment. Based on this comment, we have added a section regarding the uniqueness of this review on page 6 lines 263-268.

Added: “As an emerging area of research that has garnered significant interest but still lacks sufficient clinical evidence for definitive conclusions, further studies are needed to explore the sleep biosignatures in individuals with diabetes and the potential for sleep interventions in diabetes care and glycemic management. This article provides a concise narrative review, summarizing the currently available evidence and aiming to identify research gaps for future exploration. ”

Reviewer 2 Report

Comments and Suggestions for Authors

In this study, the author reviewed the effects of slow wave sleep (SWS) on glucose homeostasis, insulin secretion capacity, and diabetes. The authors should provide a schematic representation of the mechanisms between SWS and diabetes and obesity. Or, they should provide a schematic representation of how sleep architecture changes contribute to diabetes. Finally, please discuss genes or hormones that may be involved in the association between SWS and diabetes.

Author Response

Comments: In this study, the author reviewed the effects of slow wave sleep (SWS) on glucose homeostasis, insulin secretion capacity, and diabetes. The authors should provide a schematic representation of the mechanisms between SWS and diabetes and obesity. Or, they should provide a schematic representation of how sleep architecture changes contribute to diabetes. Finally, please discuss genes or hormones that may be involved in the association between SWS and diabetes.

Response: Thank you very much for your helpful comment. Based on this comment, we have added Figure 1 in the revised manuscript to show the main mechanisms between SWS and diabetes and obesity.

We also added a discussion regarding hormones that may be involved in the association between SWS and diabetes on page 4 lines 165-170.

Added: “Further, SWS plays a key role in modulating endocrine function. It suppresses the release of hormones from the hypothalamic-pituitary-adrenal axis, such as cortisol, while promoting the increased secretion of growth hormone and prolactin [50]. Enhancing SWS in healthy individuals through hypnotic suggestions significantly increased the release of growth hormone and prolactin, while also shifting the sympathovagal balance toward reduced sympathetic dominance [51].”